# Discussion on the Relationship between Computation, Information, Cognition, and Their Embodiment

**DOI:** 10.3390/e25020310

**Published:** 2023-02-08

**Authors:** Gordana Dodig-Crnkovic, Marcin Miłkowski

**Affiliations:** 1Department of Computer Science and Engineering, Chalmers University of Technology, 412 96 Gothenburg, Sweden; 2Division of Computer Science and Software Engineering, School of Innovation, Design and Engineering, Mälardalen University, 722 20 Västerås, Sweden; 3Institute of Philosophy and Sociology, Polish Academy of Sciences, ul. Nowy Świat 72, 00-330 Warszawa, Poland

**Keywords:** computing nature, info-computationalism, morphological computing, information physics, evolution, self-organization and autopoiesis, actors and agent networks

## Abstract

Three special issues of Entropy journal have been dedicated to the topics of “Information-Processing and Embodied, Embedded, Enactive Cognition”. They addressed morphological computing, cognitive agency, and the evolution of cognition. The contributions show the diversity of views present in the research community on the topic of computation and its relation to cognition. This paper is an attempt to elucidate current debates on computation that are central to cognitive science. It is written in the form of a dialog between two authors representing two opposed positions regarding the issue of what computation is and could be, and how it can be related to cognition. Given the different backgrounds of the two researchers, which span physics, philosophy of computing and information, cognitive science, and philosophy, we found the discussions in the form of Socratic dialogue appropriate for this multidisciplinary/cross-disciplinary conceptual analysis. We proceed as follows. First, the proponent (GDC) introduces the info-computational framework as a naturalistic model of embodied, embedded, and enacted cognition. Next, objections are raised by the critic (MM) from the point of view of the new mechanistic approach to explanation. Subsequently, the proponent and the critic provide their replies. The conclusion is that there is a fundamental role for computation, understood as information processing, in the understanding of embodied cognition.

## 1. Introduction

As a starting point, two positions with respect to cognition are introduced: cognition as a part of computing nature vs. cognition in the new mechanistic framework of explanation. First, a framework of computing nature/info-computation is presented for the analysis of cognition, natural and artificial, that enables the study of information processing/computational phenomena based on current knowledge of related research fields, such as natural/unconventional computing, bioinformatics, neuroscience, relational physics, etc. It facilitates two-way learning: from nature to the study of information-processing artifacts, and from information-processing artifacts to models and theories of natural cognitive systems. The info-computational framework requires the broadening (generalization) of involved concepts of information, computation, and cognition. The received notions are connected to the existing computational technology and do not reflect the possibility of natural computational processes and natural information as a basis of cognition. One more important step in this “naturalization of cognition” is recognizing the continuity between cognition in living organisms and biology, bringing into the picture the developmental and evolutionary view of cognition as a process in all living entities. Cognition in nature is the realization (implementation) of life, which is implying that we only can understand natural cognition in the light of evolution. In computing nature, cognition is a process of problem solving and learning that leads to increasingly complex and competent organisms.

Afterward, an opposed position is presented, which does not assume the broadening of concepts of information, computation, and cognition. In particular, the notion of computation is understood restrictively along the lines of the new mechanistic framework of explanation as primarily explanatory.

We offer two possible computational perspectives on cognition. Interestingly, for all the differences between the two positions, they converge in their acceptance of the computational framework for understanding cognition as requiring adaptive information processing recruited by the evolutionary processes for purposes of control. This common understanding underwrites the role of bodily morphologies, but also neural systems and similar dedicated biological structures.

## 2. Gordana Dodig-Crnkovic: Cognition in the Info-Computational Nature Framework

### 2.1. Definition of Terms Used in the Framework

In the info-computational formulation of cognition, based on contemporary positions in the theory of computing, physics, chemistry, biology, neuroscience, and philosophy of information and computation, generalizations of several fundamental concepts are required. New insights are reflected in generalized concepts. New concepts emerging from the state of the art of sciences relevant to cognition are listed below:

***Information** is the structure of reality—the* fabric of the universe/nature for a cognitive agent [1,2,3,4]. It is a structure consisting of differences in one system that cause difference in another system. In other words, information is observer-relative. It does not mean it is subjective. It is the same kind of observer-relativity as in the theory of relativity in physics. Information and reality are seen as one by living organisms as cognizing agents [5,6,7,8,9,10,11,12].

***Computation** is information processing* (dynamics of information) executed by the physical morphology of a substrate, computing nature, on a given level of organization, [13,14,15,16,17,18,19,20,21,22,23,24,25,26,27,28,29,30,31,32]. Both information and computation appear on different levels of organization/abstraction/resolution/granularity.

*The **Autonomous Agency** of living beings is the basis of the development and evolution of cognition in nature.* Of all autonomous agents (entities capable of spontaneously acting on their own behalf) living agents are characterized by their ability to actively make choices that increase their probability of survival. This is based on the self-organization of matter [33], as physical matter is also “active matter” and not only Newtonian “inert matter”. This activity/agency of physical/chemical/biological/cognitive matter relies on the use of energy from the environment, since living organisms are thermodynamic systems far from equilibrium, as developed by Prigogine [34].

***Cognition** comes in degrees*, and it is a process of realization of life [8,35] possessed not only by humans, but by all life forms. It is a network of networks of life-sustaining processes enabling every living organism to perceive its environment, react and adapt, [4,13,14,15,16,17,18,19,20,21,22,23,24,25,26,27,28,29,30,31,32,36].

***Evolution.*** In the info-computational model of evolution, there are several levels of organization at which evolution happens: physical (based on the agency of particles), chemical (based on the agency of molecules), and biological (based on the agency of living cells). The model of cognition in computing nature supports the Extended Evolutionary Synthesis (EES), which considers that not only random mutations, but also constructive development through sequences of changes caused by the laws of physics and chemistry (computing nature terminology: morphological computation), in an organism interacting with its environment, lead to the development of new structures which are exposed to processes of reciprocal causation with the environment and also natural selection. Proponents of the EES [37,38,39,40] argue for a new approach to evolution that does not refute Darwinian thinking, but builds further, exploring and explicating underlying mechanisms of evolutionary biology in several dimensions—*genetic*, *epigenetic*, *behavioral*, *and symbolic*, as suggested in [37]. Those dimensions are levels of organization that correspond to different levels of agency, i.e., information communication mechanisms, and thus different levels of self-organization, and have already been addressed in terms of information processing; that is, computation.

Outdated concepts, such as the belief that cognition is exclusive to humans, the idea that computation is limited to the Turing machine model, and the assumption that evolution is driven solely by random variation, and not an active agency of interacting constituents at different levels, hinder our ability to gain new understanding. Current research indicates that it is preferable to adopt updated, generalized concepts that take into account the cognitive agency of living organisms in a dynamic environment.

### 2.2. Computing Nature and Evolution of Cognition

Cognitive science has its roots in psychology and the philosophy of mind, historically focused on the human as a cognizing agent. Allowing for other, non-human cognitive agents, Piccinini [41] argues that cognitive science has transformed itself into cognitive neuroscience, and cognition can be understood as a result of neurocomputation in organisms with nervous systems, thus acknowledging neural processes as computation. Even though Piccinini goes a step beyond the conventional anthropocentric understanding of cognition, he retains neurocentrism.

Meanwhile, recent research finds that “cognitive operations we usually ascribe to brains—sensing, information processing, memory, valence, decision-making, learning, anticipation, problem-solving, generalization, and goal-directedness—are all observed in living forms that don’t have brains or even neurons” [42].

In the info-computational framework [43], building on contemporary results from biology and basal cognition (cognition of a single cell), cognition is generalized a step further from neurocentrism, to include all living forms, not only those with nervous systems. For example, symbolic, language-form information processing, typical of human communication, also exists on the level of chemical languages used by bacteria in bacterial quorum sensing, as Bassler, Ben-Jacob, and others have described, [4,5,6,7,8,9]. Bacteria cooperatively collect “latent information from the environment and other organisms, process the information, develop common knowledge, and thus learn from past experience” [44,45,46,47,48].

Plants as well can be said to possess basic cognitive functions such as memory (traces of past events in their bodily structures), on which their behavior is based, and the ability to learn (adapt and change their morphology) as a result of interactions with the environment, which are rudimentary forms of cognition [49].

In the info-computational approach to cognition, evolution is understood in the EES sense [37,38,40,50], which emphasizes *constructive development* (not only random variation) and *reciprocal causation* (not only selection). Evolution is a result of interactions (information communication) between living agents [51,52,53], cells, and their groups on different levels of organization, as well as intrinsic information processing—which is computation. McMillen, Walker, and Levin show explicitly how information theory can be used as an experimental tool for integrating disparate biophysical signaling modules [54], which are essential for evolution.

The Computing Nature (Naturalist computationalism) framework makes it possible to describe all cognizing agents (living organisms and artificial cognitive systems) as informational structures with computational dynamics, [13,14,15,16,17,18,19,20,21,22,23,24,25,26,27,28,29,30,31,32]. Computation is manifest in changing the morphology of a physical body.

It should be added that neurocentrism not only neglects the evolutionary and developmental side of cognition, but it also disregards the cognitive role of the somatic cells in organisms with nervous systems—thus abstracting away embodiment. However, it has been shown, for example, that the immune system plays a decisive role in the cognitive behavior of organisms by distinguishing the “self” from the “non-self”, and that it is in constant interplay with the nervous system and the rest of the body [55].

The advantage of the info-computational approach to cognition is that it is capable of modeling complex behaviors, adaptation, evolution, and learning as found in nature. Cognitive computing and cognitive robotics are attempts to construct abiotic systems exhibiting similar cognitive characteristics to natural systems. Since cognition in nature comes in degrees, from basal cognition upwards, it is meaningful to talk about the cognitive capabilities of artifacts that come in degrees, even though cognition for an artifact does not serve to assure their existence or reproduction, which is the primary role of cognition in biotic systems, at least on the cell level. Of course, cognition in complex cognitive systems such as humans involves many more aspects than survival.

Given the rapidly progressing development of increasingly sophisticated, artificially intelligent cognitive computational systems, a framework that can seamlessly connect the natural with the artificial is useful for learning in both directions—from the natural system to the artificial and back [56].

### 2.3. Information, Computation, Cognition: An Eternal Golden Braid

Life can be analyzed as cognitive processes unfolding in a layered structure of nested information network hierarchies with corresponding computational dynamics (information processes) in nature—from the molecular to the cellular, organismic, and social levels.

To describe life as a cognitive process, we introduced two fundamental theories about the nature of the universe and their synthesis.

The first one with *a focus on processes* is the idea of the computing universe (naturalist computationalism/pancomputationalism) in which a cognizing agent sees the dynamics of physical states in nature as information processing (natural computation) [57,58,59,60,61,62].

Computation is, in general, information processing. A suitable model for computation within an info-computational framework is Hewitt’s Actor model. Hewitt’s actors can be seen as autonomous agents in this context, [63,64,65]. Here, an autonomous agent is defined in the sense of Kaufman [66] as an entity capable of acting on its own behalf.

The complementary fundamental theory with a focus on structures is informational structural realism, which takes information to be the fabric of the universe. Whatever exists for an agent comes in the form of information. The world presents potential information for an agent. Information is relational, according to Floridi and Sayre [1,2].

Info-computationalism is a synthesis of informational structural realism (nature is an informational structure for an agent) and natural computationalism/pancomputationalism (nature computes its future states from its past states). It is also a variety of physicalism, where physical matter is represented by information (for an agent), and information processing is physical computation.

The concept of information from the point of view of computing nature, as it appears at different levels of organization, is given in “Information, Computation, Cognition. Agency-Based Hierarchies of Levels” [67]. In order to understand agency in the world, we have to understand the constructive mechanisms that connect living beings with inanimate nature. In his book [68], Deacon provides a good account of a hierarchical organization of a biological cognizing agent, starting with abiogenesis through self-organization of biological structures, which are used as building blocks in the subsequent construction of increasingly complex living organisms. Deacon proposes the framework for information processing in living systems that distinguishes between the following three levels of natural information (for an agent):(Shannon) (data, pattern, signal) (data communication)—the level of syntax(Shannon + Boltzmann) (intentionality, aboutness, reference, representation, relation to object or referent)—the level of semantics and((Shannon + Boltzmann) + Darwin) (function, interpretation, use, practical consequence)—the level of pragmatics.

Three types of information can be seen as stages in the development from matter to mind.

Deacon’s three levels of information organization parallel his three formative mechanisms: 

{Mass-energetic {Self-organization {Self-preservation (semiotic)}}}

with corresponding levels of emergent dynamics: 

{Thermo- {Morpho- {Teleo-dynamics}}}

and parallel Aristotle’s causes:

{Efficient cause {Formal cause {Final cause}}}

This notation follows Salthe [69], where in {stage 1 {stage 2 {stage 3}}}, Stage 2 develops out of Stage 1, and Stage 3 develops from Stage 2.

Deacon’s model of information organization levels shows the fundamentally embodied character of information through its physical manifestations.

The dynamic of information (information processing, computation) underlying the emergence of three levels of self-organization and autopoiesis can be represented in terms of agent-based models, such as Hewitt’s actor model of computation [63].

### 2.4. Old Computationalism: The Turing Machine Model of Abstract (Logical) Information Processing

It has been realized already twenty years ago by Scheutz and Sloman that computationalism today is not what it used to be, namely the thesis that cognitive computation can be adequately modeled as a Turing machine, [70,71].

The Turing Machine following a given algorithm may only be used for the description of certain aspects of the functioning of living organisms. However, we need computational models for the basic characteristics of life: the ability to differentiate and synthesize information, make a choice, adapt, evolve, and learn in an unpredictable world. That requires computational models which are not simply sequential symbol manipulation, predefined and by definition provided with potentially unbounded resources. Cognition in nature is about survival with finite resources. As Siegelmann aptly argues:

*“Biological processes are often compared to computation and modeled on the Universal Turing Machine. While many systems or aspects of systems can be well described in this manner, Turing computation can only compute what it has been programmed for. It cannot learn or adapt to new situations. Yet, adaptation, choice, and learning are all hallmarks of living organisms”*.[72]

### 2.5. New Computationalism: Physical Levels of Embodied Distributed and Concurrent Computation—Morphological Computation

Morphological computation in the info-computational framework is a process of creation of new informational structures, as it appears in nature, living as non-living. It is a process of morphogenesis, which, in biological systems, drives development and evolution [73,74,75].

If the whole of nature computes, this computation happens on many levels of organization of the physical matter. In [76], three generality levels are introduced by Dodig-Crnkovic and Burgin, with computation defined as:***Any******transformation of information** and/or information representation*. That leads to natural computationalism in its most general form.***A******discrete transformation of information** and/or information representation*. That leads to natural computationalism in the Zuse and Wolfram form, with discrete automata as a basis.***Symbol manipulation***. That leads to the Turing model of computation and its equivalents.

The current state of the art on typologies of computation and computational models is presented in [77,78,79], outlining a basic structural framework of computation. More about bodily information processing and the role of morphological computation can be found in [80,81] as well as in the review article [82] addressing the recent progress in the understanding of the embodiment of computing systems.

### 2.6. Actor Model of Morphological Computation

Looking at cognition on the cell level, we observe that the cell consists of a number of functional parts which interact and exchange information. Interacting parts can be modeled as autonomous agents (actors), and information exchanges as computation.

“In the Actor Model [Hewitt, Bishop and Steiger 1973; Hewitt 2010], computation is conceived as distributed in space, where computational devices communicate asynchronously, and *the entire computation is not in any well-defined state.*”, [63]. Hewitt’s “computational devices” are computational autonomous agents—informational structures capable of acting on their own behalf.

In contrast to other models of computation that are based on mathematical logic, set theory, algebra, etc. the Actor model is based on physics, [65]. That makes them especially suitable for modeling morphological computation.

### 2.7. Natural Information: Agent-Based and Relative

The info-computational framework adopts a combination of information definitions by Bateson [83] and Hewitt [84]. Bateson’s definition is:

*“Information is a difference that makes a difference”*.[83]

Note that Aaron Solman’s article “What’s information, for an organism or intelligent machine? How can a machine or organism mean?” [39], makes it clear that Bateson by this definition actually refers to data as atoms of information.

Hewitt’s definition is:

*“Information expresses the fact that a system is in a certain configuration that is correlated to the configuration of another system. Any physical system may contain information about another physical system”*.[84]

This together gives us the following definition:


*Information is defined as the difference in one physical system that makes a difference in another physical system.*


It implies the relational character of information and thus agent-dependency in the agent-based or actor model [63,64,65].

As a synthesis of informational structural realism and natural computationalism, the info-computational framework adopts two basic concepts: information (as a structure) and computation (as a dynamics of an informational structure), [15,85].

In consequence, the process of dynamic changes in the universe makes the universe a huge computational network of networks, where computation is information processing [86]. Information and computation are two basic and inseparable elements necessary for naturalizing cognition [27]. It is also necessary to keep in mind that there is no information without physical implementation, as argued by Landauer in ”The physical nature of information” [87]. Thus, each information–computation level has its corresponding matter–energy basis.

### 2.8. Agency-Based Hierarchies of Levels for an Autonomous Agent

A living agent is a special kind of actor “that can reproduce and is capable of undergoing at least one thermodynamic work cycle” [88].

Although a detailed physical account of the agent’s capacity to perform work cycles and so persist in the world is central for the understanding of life/cognition, as [66] and [89] have argued in detail, here we are primarily interested in the info-computational aspects of life and not in its energetic/metabolic side. That means that we take information and computation as the basic building block concepts, corresponding to *structure* and *process*, or *being* and *becoming*. We tacitly assume the existence of the physical world, on which structures and their dynamics are implemented.

Given that there is no information without physical implementation [87], computation as the dynamics of information is the execution of physical laws.

Kauffman’s concept of autonomous agency (also adopted by Deacon) suggests the possibility that life can be *derived* from physics (via chemistry). That is not the same as the claim that life can be *reduced* to physics, which is false.

We witness the emergence of information physics [90,91] as a possible reformulation of physics that may bring formulation of physics and life/cognition closer to each other. This development smoothly connects to the info-computational understanding of nature [92].

The reality for an agent is an observer-dependent reality [93,94] based on available information from the environment and the agent’s capacities to identify and process that information.

### 2.9. Morphological Computation on a Cellular Level: Somatic Bio-Computation as Dynamic of Cellular Information

The work of Levin suggests a broad range of applications for nature-inspired cognitive information-processing architectures based on biological cognition connecting genetic networks, cytoskeleton, neural networks, tissues/organs, and the organism with the group (social) levels of information processing, [42,75,95,96,97]. They provide a smooth connection between the deepest levels of biological mechanisms and high-level cognition.

Levin et al. [95,96,97] show how biology has been computing through somatic memory (information storage) and biocomputation/decision making in pre-neural bioelectric networks before the development of neurons and brains. He makes a very important connection between ordinary cells and neurons, revealing their evolutionary connections and common information-processing mechanisms. 

Such insights from bio-cognition can help the development of new AI platforms, applications in targeted drug delivery, regenerative medicine and cancer therapy, nanotechnology, synthetic biology, artificial life, and much more.

The info-computational framework is treating cognition as an open-ended process of spontaneous behavior of matter, such as self-organization where computation, for the most part, proceeds as signal processing in natural systems, and only under special circumstances takes the form of symbol manipulation and language-based communication, [98,99].

Mechanisms of cognition, based on natural computation [100]/morphological computation [101,102] are far more sophisticated than the machine-like classical computationalist models based on abstract symbol manipulation [103]. They conform to the view that rule-based machines are not good enough models of natural cognition of highly complex living organisms [104,105].

Embodiment is the fundamental characteristic of cognition, which then implies that manifestations of embodiment such as senses, feelings, and emotions must be considered constitutive of cognition, [106,107,108,109]. In the framework of computational nature, mechanisms driving cognitive processes are smoothly connected and the difference between various levels of cognition is in the complexity of networks of autonomous agents which constitute them. They are molecules in the case of cells; cells in the case of organs; and organisms in the case of social groups.

The info-computational approach builds on the current scientific knowledge about processes in nature, translating them into the language of natural info-computation. It increases understanding of cognitive processes in various types of agents, biological and synthetic [110].

## 3. Marcin Miłkowski: Reply to “Cognition in the Info-Computational Nature Framework”

### 3.1. Computation? Yes! But Not Just Everywhere

While we share the naturalistic perspective on natural computation and cognition, my account of physical computation [111] diverges in several ways from the info-computationalism defended by GDC:(1)I do not share the idea that the physical is best understood in terms of information.(2)My account of computational mechanism is more constrained than that of information processing.(3)I take the notion of cognition to be highly context-dependent (or relative to a theory).

I shall elucidate all three points below.

### 3.2. Natural Information Is Everywhere

While it is undeniable that the notion of information can be understood in terms of a difference that makes a difference to another physical process, as GDC does, there are fruitful applications of this concept and somewhat stretched overextensions.

The concern that a concept is applied too widely is at the core of many philosophical debates. For example, one may worry that the notion of cognitive representation is used too liberally [112]. Alternatively, one may worry that if anything could be truly described in computational terms, then the notion of computation would be entirely trivialized [113,114].

The overall argument structure in these instances seems to be as follows:Concept C is defined to cover all its referents but also to instances intuitively understood as exemplary non-Cs, which means that C is overextended.If concept C is overextended, then its definition is too wide.If the definition of C is too wide, then the use of C to distinguish between specific relevant instances of C and cases in which C is not present is impossible.

Thus, overextending concept C leads to trivializing C.

For example, if the concept “cognitive representation” covers any causal mediator of behavior, then representational explanations cannot be distinguished from any explanation of behavior, rendering the notion of representation trivial [112]. By learning that there is a cognitive representation involved in a given behavior, we do not obtain any new information. In other words, calling something “cognitive representation” would not make it in any way different from other causal mediators of behavior. The notion would lose specificity and become entirely redundant. Similarly, if any physical process is computational, then we cannot ascribe any unique features to physical computation because they would be shared by any physical process. There would be no way to ascribe any specifically computational property to those systems that we intuitively call ‘computational’.

This is, however, not a decisive blow against any attempt to extend the scope of a given concept. Obviously, commonsensical concepts are redefined to become more precise or theoretically useful. It is quite clear that before Turing’s groundbreaking work [115], the notion of computation was meant to refer only to human operations on numbers. Afterward, and not without the vehement opposition from philosophers of language such as Ludwig Wittgenstein [116], it was fruitfully extended. The question is, nonetheless, whether “computation” or “information” is among these few concepts that should be ascribed universally to anything in spacetime.

In traditional philosophical terminology, this would make information and computation alike to a transcendental, just like truth, goodness, and beauty. For spacetime beings, we can talk of spacetime locations as universally applicable, which does not render the notion of spacetime location trivial at all. As I argued elsewhere, the claim that computation is physically universal (sometimes dubbed ‘pancomputationalism’, but GDC insists that we should call it ‘info-computationalism’) need not render “computation” trivial [117]. The real threat of trivialization would occur, *were any computation justifiably ascribable to any physical process*. This is similar to the notion of “spacetime location”: if you could ascribe any physical spacetime location to any physical event or process, the notion of “spacetime location” would be utterly meaningless. However, this is not what defenders of info-computationalism claim.

The question is whether the extended usage is innocuous. Indeed, it is arguable that “information” is applicable to most physical events (if they have any effects or at least correlate with any other events), and would be applicable to a large number of possible worlds. However, this does not seem to worry anyone. The worry is only that there are extreme versions that make information the basic stuff of the universe, which seems to confuse the stuff of which vehicles of information are made with information itself. While you could defend the ontology of “it from bit”, it seems to flout the precept of ontological seriousness [118]: how can information be its own vehicle? Contra defenders of information as the basic stuff of the universe (including such classics as [119]), we can simply state that it is much more intelligible that there are physical vehicles of information. 

At the same time, there are various ways one can understand the ontological commitments of our best physical theories. Scientific realists can embrace structural realism about the physical, e.g., [120], but that does not imply that all there is, are informational structures; it implies that there are structures, which can, but need not be, interpreted in informational terms.

However, in contrast to scientific realism, which seems committed to physical vehicles of information, non-realist positions are available. One could be instrumentalist about information and claim that it is merely a useful notion. For one, I do not find this enlightening, if not for the fact that instrumentalism owes us a substantial account of the utility in question, which usually seems to be posited without answering the basic question: What makes a notion useful, if it has no referential credentials at all? Why are some non-referring concepts useful and some useless? A realist can easily say a notion is not useful unless it refers to what is out there (even if it is an idealized notion, for idealizations can be made compatible with realism in one way or another [121,122]).

Another option available to defenders of informational ontology is constructivism, according to which anything that exists, exists only in the eye of beholder. It is difficult to state this position without the air of the paradox (what if beholders do not exist?), but a more careful analysis can dispel some of it. What is at stake, in most cases, is that entities in constructivist ontology are taken to be response dependent in a lovely fashion. Let me elaborate: Daniel Dennett [123] introduced a distinction between *lovely* and *suspect* properties: the former ones are merely dispositional properties that a class of observers might respond to, the latter ones are properties that at least some observers have responded to (you cannot be suspect without anyone suspecting you). That would imply that the ontological commitments of informational constructivism are heavily dispositional: existence is taken to be dependent on the dispositions of observers (receivers of information?) to respond to what there is.

Frankly, I find such ontology confusing. 

While it is certainly true that any naturalist account of ontology should consider our best theories and their ontological commitments, which makes ontology reasonably dependent on our epistemology, we need not understand that what there is depends on whether it could be theorized or thought about, even in a dispositional fashion. 

This is simply too convoluted a way to think of reality, in particular because some theoretical entities might be defined or introduced conceptually in a fashion that obviates the possibility of responding to such entities (think of the mere conceptual possibility that there are black holes disallowing the flow of information from them). 

Moreover, even though constructivism is inspired by Kantian transcendental idealism, it seems to rely on the same flawed argument, which has been long criticized as invalid. The original argument goes roughly like this: we know things only as they are related to us (or as phenomena), so we cannot know things as they are in themselves. The only knowledge we can have is about phenomena, and we must remain agnostic about the things in themselves. However, this conclusion cannot follow from the premise, unless you stipulate that knowledge of phenomena is not knowledge of things in themselves. However, you cannot stipulate that, on pain of begging the question. Similarly, even though any knowledge is constructed, and our cognitive architectures impose constraints on what and how we can know, it does not imply anything about the nature of the world as it is. On the contrary, given that ‘know’ and similar verbs are understood as factive, you cannot know anything falsely. In the modern formulation, even if all agents are capable of responding to is information that makes difference to them (which is trivially true), it does not imply that the basic reality is merely information. Take a simple example: when I count apples, all I can obtain as a result is a natural number. This, however, does not imply that apples do not exist, or that numbers are all that there is, just because I can count.

Hence, while the notion of information may as well be transcendental or universal, it does not make it a basic stuff of the universe, at least not more than truth is.

### 3.3. Computation versus Information Processing

However, the ontology of information is much less worrisome to most than pancomputationalism, in particular in its limited versions. Notice that some of the philosophical accounts of physical computation have the implication that any causal process implements a computation [124]. Again, this bullet can be bitten by computationalists. Why oppose it, then?

My strategy to answer this question was epistemological. Given that my account of computational implementation relies on the mechanistic account of explanation [111] the criteria used to evaluate whether an attempt to ascribe computational properties to a certain physical process is justified or not are mostly explanatory. While, following Chalmers [124], one can redescribe any causal process in computational terms, such redescription does not bring much insight in itself, unless there is some specific cognitive purchase: predictive, explanatory, or related to control. 

In the mechanistic framework, it is admissible to posit mechanisms only as responsible for specific phenomena. For example, to explain how an ATM allows me to withdraw money from my account by providing me with paper bills (an observable phenomenon), I can posit a complex computational mechanism that controls the machinery for distributing paper bills when certain conditions are met (such as inserting a valid card and entering a secret PIN). Additionally, our understanding of physical phenomena is enhanced when we can speculate about what would happen if certain conditions were different, such as entering a false PIN at an ATM. In contrast, the idea that all causal structures are computational in some sense may not be useful for explanation because we may have no understanding of the *specific* phenomena to which they contribute. Some observable data may never be worthy of theorizing in terms of phenomena, such as the fact that I have lost a certain number of socks in my life until now [125].

The use of computational modeling in explanation goes beyond simply describing observed data. For example, it allows us to consider counterfactual scenarios, such as why an ATM provided more banknotes than requested. However, if a computational model merely restates the existing knowledge about the physical system, there is little reason to utilize it.

These are, then, two basic epistemological reasons why I would oppose even a limited pancomputationalism: making it transcendental is problematic because we cannot specify the phenomena for all physical causal structures, and even if we can, sometimes it brings no explanatory power. How does my account of computation constrain the notion so as to be less liberal than the one defended by Chalmers, but still admit of legitimate scientific uses though? There are two considerations at play [111].

First, there are specific scientific usages of the notion of computation that are admittedly broad. For example, one might want to estimate the upper bounds of the computational power of the whole physical universe [126]. For such uses of the notion, I recommend using the expression “information processing”, which is often used interchangeably. However, since we have two distinct terms, we can say that not all information processing occurs in computational mechanisms. In this particular example, we might treat the entire universe as a gigantic information-processing mechanism to establish its physical computational limits. Even critics of pancomputationalism should admit that there is nothing particularly wrong with doing this. On the contrary, such limits may be essential to our understanding the limits of tractability.

Second, similarly to Piccinini [127], see also [128], I focus on functional mechanisms of computational phenomena. The motivation is simple: in scientific explanations, we are mostly interested in naturally evolved (usually biologically functional) or engineered computational mechanisms. In both cases, these are mechanisms that are capable of failing. While causal structures per se cannot fail, functional mechanisms can [129]. Now, the question is what notion of teleological function suits these cases most. In contrast to other proponents of the mechanistic account of computation, such as Piccinini [127] or Dewhurst [130], I rely neither on proper biological functions, e.g., [131] nor on functions understood as roles in a complex system [132]. Instead, I focus on the design of the computational mechanisms: these are designed to compute. Following the account defended at length by Ulrich Krohs [133], there are two conditions that functional mechanisms should satisfy:(1)The parts of the mechanism should have been selected as types (and not as particular tokens) according to a certain design. Think of DNA involved in folding proteins: it does not specify proteins as individuals (tokens) but as types only.(2)The selection process should be causally relevant to the existence of the mechanism.

Note that there is no conceivable mechanism that selects parts of the whole universe to compute anything. However, there are mechanisms (and design histories) involved in how people produced analog computers, and digital machines, and in how evolution endowed nervous systems with their capacities to compute. Krohs’s notion makes it relatively easy to test whether something is a function or not: just look at the selection history. (This obviously makes me nod in agreement when cognition is understood in computational terms due to evolutionary reasons, as the computing nature framework implies.)

This way, the notion of computational mechanism can be restricted for explanatory purposes. At the same time, it is sufficiently liberal to cover unconventional computation: in my account, I do not presuppose that there is only one and unique mathematical account of computation (such as the Universal Turing Machine), endorsing the principle of transparent computationalism [134]. In this respect, I fully endorse GDC’s broad characterization of computation.

### 3.4. The Context-Dependent Notion of Cognition

Let me bring these points together. There are excellent reasons to believe that functional computational mechanisms are in place in biological agents; as these agents act adaptively, their adaptive control of behavior must rely on information processing. Evolution endowed biological agents with cognitive processes for that very reason. While some defenders of computationalism did claim that computation is sufficient for cognition [135,136], a more modest claim is that the notion of cognition involved in cognitive science explanations requires computational mechanisms. I have not encountered any plausible arguments to the contrary yet [137].

At the same time, notice the modesty of the claim: it is limited to cognitive (neuro)science as engaged in explaining adaptive control of behavior in various ways, and there might be some other ways of conceiving cognition. Still, the notion is probably so broad as to encompass possible cognitive processes in life sciences, such as in bacteria, fungi, or plants, because they also display adaptive behavior. Nevertheless, other fields of scientific inquiry might have reasons to conceive of cognition in some other way, or restrict it even more (for example, by requiring that it relies on intrinsic mental representation, or depends on skilled intentionality, or social institutions or language; take your pick). For this reason, I endorse a limited form of pluralism about cognition: depending on your field of inquiry or the particular context of a certain theory or explanation, your notion of cognition will inevitably vary somehow. However, as long as it involves the adaptive control of behavior, cognition is computational.

## 4. Gordana Dodig-Crnkovic: Answers to Criticism of Marcin Miłkowski towards the Computing Nature as an Approach to Framing Cognition in Terms of Information and Computation

### 4.1. Computation? Yes! And Why Not Everywhere? Matter-Energy Is Everywhere; Space-Time as Well

The question of whether computation and information are among these few concepts “that should be ascribed universally to anything in spacetime” in computing nature is answered simply by: yes, of course!

Information and computation are not like truth, goodness, and beauty. They are like matter–energy! Information is as “matter” and computation as “energy”. In a similar way, as the notion of space/time is not trivial, matter/energy is not trivial, and information/computation is not trivial.

Matter/energy in space/time are fundamental notions of physics, while truth, goodness, or beauty are high-level complex concepts that are context-dependent and culturally determined. We have internationally accepted units for mass, energy, length, and time, but there are no such units for beauty or truth. For a good reason. Computing nature does not ‘stretch’ or ’overgeneralize’ concepts of information and computation more than physics stretches notions of matter and energy, and that level of generality need not render computation trivial [117]. 

The real threat of trivialization would occur, *were any computation truly ascribable to any physical process.* That would make the universe a “gray goo”. I agree that in that case, trivialization indeed happens. Such mapping is non-physical and nonsensical.

### 4.2. Materiality of the Info-Computational Universe: Physical Computation, Morphological Computation

The worry that making information the basic stuff of the universe confuses the stuff of which vehicles of information are made with the information itself might sound as a legitimate worry. While one could defend the ontology of “it from bit” on epistemological grounds, the question is: *how can information be its own vehicle?* Important question! It has an answer. Information is not its own carrier; it is carried by physical signals, and it is always implemented in physical substrate.

Let us start from the definition. Information is defined as a difference in one system that makes a difference in another system. Everything in the world (umwelt for a living organism) that affects a cognizing agent comes as input that is registered by its sensors, for which it is *information*. It is not only a photon of light or a chemical molecule, odor, or another kind of interaction mediator that sensors register, but it is for an organism (or in general an autonomous agent) a sign or data or *information about the world which exists for that agent.*

There is no information without physical implementation [87], as pointed out already. Information always comes through some signal which is physical, has energy, mass, or both. An agent is a material object as well, made of matter-energy. *All agents and all the world have that fundamental materiality*. However, that is not the focus in the framework of informational universe/computing nature. The materiality is assumed, and it is not the aspect we address. The informational universe is built of information (for an agent). Even the energy of a signal that an agent receives is information for an agent.

In the framework where information and computation are used to trace and affect events in the real world, materiality is a canvas on which the painting of reality is drawn for an agent.

There is a parallel with virtual reality. We have a computer that runs a game, but for the reality of the game itself, it does not matter. What matters for the players are the internal rules of the game.

For scientific realists [120], reality consists of informational structures; it implies that there are *structures*, which can but need not be interpreted in terms of their underlying materiality.

Positions of instrumentalism are useful as they reflect attitudes of the informational era and have practical value.

### 4.3. In Defense of Constructivism

Constructivism “*according to which anything that exists, exists only in the eye of beholder*” is radical constructivism, and it should be understood in its natural context, which is the theory of learning.

Subjective perception is not the main focus of constructivists, but instead *the process of construction of knowledge*. That knowledge is always constructed by/in a cognizing agent.

The articles [138,139,140] give the details of my position on the topic of reality/knowledge construction for a cognizing agent through the processes of morphological computation in info-computational nature.

Physicist Heinz von Foerster was one of the leading constructivists of his time. He has written a ground-breaking book “Understanding Understanding” [141]. As a physicist, he of course did not believe that physics and the very existence of the world depend on our personal dispositions, or that “*anything that exists, exists only in the eye of beholder*”. Science is based on the reproducibility of experiments which would be impossible if the experiment would (only!) be in the eye of beholder, and would not independently *exist* in the physical world. 

However, what von Foerster and constructivism says is that *the way an agent perceives the world is dependent on the agent’s cognitive architecture*. Humans and fish do not perceive the world in the same way. That is a completely different and rather natural position, based on a scientific understanding of how cognition works in biology and neuroscience, among others.


*“Indeed, naturalist account of ontology looks at our best theories and their ontological commitments, which makes ontology reasonably dependent on our epistemology, we need not understand that what there is, depends on whether it could be theorized or thought about, even in a dispositional fashion.”*
(MM)

Von Foerster agrees with this [141]. The black hole example is instructive. Physicists describe what can be observed, measured, and inferred about black holes. An agent observing a black hole will be able to know about it through the information that is reachable from its epistemological horizon. An intelligent agent can hypothesize things that it cannot observe and, in the next step, test the hypothesis. Reasoning is also information processing, based on memory and logic (induction, deduction, abduction).

In short, it is not a coincidence that the majority of “it from bit”-like ontologies have been proposed by physicists. They presuppose the existence of the physical world and knowledge generation by an empirical and theoretical examination of the existing world. They translate physics into the language of information and computation [90,91]. That is justified and coherent. However, you may say that they do not say everything that can be said about the universe. In the virtual reality example, they do not say anything about the machine on which the simulation is run. Can we know anything about that machine?

Yes, we can, but whatever we can know, for us, is information, or informational structures (knowledge).

“While the notion of information may as well be transcendental, it does not make it a basic stuff of the universe”, (MM). As mentioned before, from the example of information physics, a fair comparison of information/computation is with matter/energy (both pairs in a sense structure/process or being/becoming). They are by no means comparable with such high-level concepts as truth and beauty.

### 4.4. Computation versus Information Processing: Sense-Making


*“While, following Chalmers [124], one can redescribe any causal process in computational terms, such redescription does not bring much insight in itself unless there is some cognitive purchase: predictive, explanatory, or related to control.”*
(MM)

Indeed, there is a “cognitive purchase”. There is always a cognitive gain, predictive, explanatory, or related to control.

Projects, driven by physicists, that are translating physics into the language of information attain cognitive benefits. See two special issues dedicated to that topic, [142,143] addressing “Information and Energy/Matter” and “Physics of Information”.


*“In the mechanistic framework, it is essential that mechanisms are posited as mechanisms responsible for particular phenomena.”*
(MM)

Informational structural realism and “it from bit” explain mechanisms. They reformulate physics in terms of, respectively, information and computation and use empirical knowledge of mechanisms as a background.

Reformulation of a statement in one language that is less accessible, to another language that is more comprehensible and manageable, is a method often used in mathematics and physics. A simple example of such a transformation is changing the coordinate system from Cartesian to polar coordinates. A problem that is nearly impossible to solve in one can be trivial in the other. Fourier transform is another example where you just translate from one formulation to another, and you solve your problem much more easily. Translating physics to information and computation makes things directly interpretable or executable by computers. Translation as a method used to increase insight should not be underestimated.

Computing nature (pancomputationalism) says just a simple thing: whatever can be said in the language(s) of physics (biology, chemistry, cognitive science, neuroscience) can be said in the language of information and computation. This goes not only causal regularities but also statistical ones.

The limitation of MM’s argument against computing nature, as well as similar ones proposed by other philosophers, comes from the fixed traditional notion of computation of the Turing Machine or the finite automata type, such as the ATM. Computation can be more and is already more today. The Internet is not a Turing Machine, it is a distributed parallel asynchronous computation network.

Computation is in general a concurrent network of processes. In biological systems, including our brains, morphological, embodied computation on several levels of organization (scales) drives structural changes. The Turing Machine is not an adequate model for such computation.

The claim that natural computation (morphological computation) is nothing but a physical computation [102] is correct.

For a naturalist, in the physical world (physics, chemistry, biology), cognition is a natural/physical process, a virtual machine running on biological hardware [71,144,145].

However, computing nature is not limited to functional computation.

Functional computation, like all the designed and engineered computations, is of course natural computation as well, but nature computes more. It “computes” or spontaneously unfolds its physical systems, which for an observer are informational structures. As Chaitin says, the universe is computing its next state by simply executing its own physics over its existing states, [61,62]. In Hewitt’s model of computation, you do not need external input for computation to execute in a system. A network of physical (chemical, biological, cognitive) agents computes by exchanging internal information (implemented in exchanges of matter–energy). The computing universe is a coherent idea with Hewitt’s actor model of distributed concurrent computation on the hierarchy of levels of organization. Natural computation is not expected to halt.

### 4.5. The Context-Dependent Notion of Cognition? Yes, Naturally: Unlike the Abstract Turing Machine Model, Morphological Computation Is Embodied, Embedded, and Context-Dependent Natural Process

Certainly, cognition is context dependent by construction in the info-computationalist framework, which is embodied, embedded, and enacted. Context independence as well as substrate independence is a feature of the old Turing-Machine-based computationalism, criticized already by, among others, Scheutz and Sloman [70,71]. Context independence is not a general feature of natural computing.

Here, for the explication of context-dependent information processing in cells and tissues as well as organisms such as planarians, I refer to the work of Levin [42,54,75,146,147], whose research studied mechanisms of signal (information) processing that exist in neurons, but also in somatic cells. As already mentioned, there is a whole new research field of basal cognition that shows how a single cell registers and processes, stores and communicates information, in order to act meaningfully (adaptively and autonomously) in a dynamic environment. All of this can be modeled as morphological information processing–morphological computation.

## 5. Marcin Miłkowski: Concluding Remarks

Thank you for bringing the analogy of the pervasive applicability of the concept of ‘information/computation’ with that of ‘matter/energy’. Indeed, we have measures for both. However, there is one crucial difference. The (arbitrary) units used to measure energy etc. are not the same as energy. The numbers we use to count apples are not apples. Information structures in the physical world are not the physical world; these are only mathematical structures inherent in the world, just like numbers.

Now, there are various ways one could understand mathematical properties of the physical. Nominalists find them problematic and in need of further elucidation. I do not see any reason for this, as mathematical properties are frequently quantified over in our best empirical theories, which makes us committed to their existence—and Aristotelian realism about properties is enough. However, these properties are properties of something, and that something is, most of the time, some kind of physical process. I am not ready to go full Platonic about these properties, not even if evolution or embodiment are cited as decisive factors for endorsing the framework.

However, there is a lot that we do converge on. Let me summarize:

In this paper, we have explored the role of information and computation in both physical and cognitive theories. We both agree that cognition requires information processing for adaptive control of behavior. Brains and nervous systems are not the only physical systems capable of cognition. In fact, even basal cognition requires sophisticated models of computation. These are explained by cognitive mechanisms, which are functional computational mechanisms because of their evolutionary underpinnings. In particular, morphological computation is one kind of physical computation that evolution found to enhance cognition.

All this seems to imply that even the positions of two co-authors on the admissible scope of the notion of “computation” diverge, there is already a lot that a naturalist position on cognition should readily embrace. The conclusion is that embodied cognition can be better understood through the lens of computation, specifically as a process of information processing [148].

## Data Availability

Not applicable.

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
