# Peer review of "Discussion on the Relationship between Computation, Information, Cognition, and Their Embodiment"

_entropy, 2023, doi:10.3390/e25020310_

Round 1
Reviewer 1 Report (New Reviewer)
This is an interesting, though purely philosophical article. It is written in the style thesis-reply-synthesis where one author (GDC) first outlines here thesis of info-computationalism and then the second author (MM) gives a critical reply. After the reply, GDC synthesizes her thoughts in light of this exchange, while MM concludes the article briefly afterward.
Formal remarks:
In principle, I think this format is a very interesting one. It reads a little like a productive reviewer’s exchange, with GDC having submitted a perspective and MM’s reply being the critical review, yet the reader might get the impression that it is a way of “recycling” a previous submission. In addition, the structure of the paper might occlude the collaborative nature (I assume this to be the case?) of the paper a bit. I felt sometimes a bit lost, given the complex structure, and I had the impression that it is mainly about GDC’s thesis (which took the main 5 chapters) -- only one chapter for MM’s reply.
I would thus suggest to re-structure it a bit
1. Introduction
2. GDC thesis, 2.1 – 2.5 (is it necessary to bring a summary at the beginning of an 8-page perspective?)
3. MM critiques
4. GDC synthesis
5. MM concluding remarks
Content
Throughout the article, a broad perspective on computing as widespread (a “naturalization of cognition”) vs. a narrow account of computing as a specific mechanistic process is presented. Since both authors subscribe to a physicalist/naturalist viewpoint, it is not very surprising that they end up noting several commonalities between their approaches, i.e. the way the paper is presented initially (as a kind of argument) is a bit misleading. In light of this physicalist hegemony, it is, therefore, more interesting to focus on the differences:
Summary of the argument: I did not see an argument here (argument for what?). Neither is it a summary of what is to come. But perhaps, it is a summary of what the author believes.
For example, GDC states that “information is the structure of reality”. This is either trivial (everything could be interpreted in terms of information) or very speculative (in particular, the claim that this does not entail a fundamental notion of subjectivity), but it is not argued for. If one accepted this statement, the remaining points are not contested: “information processing” as morphological computation (if everything is information, then every physical dynamics changes the information content, hence “computes” -- but see MM objection #(2)), “cognition everywhere” (if cognition = computation, this trivially follows). In particular, it did not become clear to me what would follow from adopting the info-computational standpoint. (it might be a good philosophical interpretation, but so what?) Also, not sure how autonomous agency makes a difference for the kinds of computations that we see in nature, nor what the “computation is ubiquitous” entails for the EES. (It seems EES is just fine w/o appeal to info-computationalism? Maybe it is consistent with is though. This is not disputed.)
Specific questions
Sec 4. “Old computationalism”
Not sure what GDC wants to say here. Do you mean that computation is not enough or do you mean computation as standardly conceived (by the UTM) is not enough but computation in the wider sense is actually enough (but why is then still “computation”)? i.e. do you claim that there are computations beyond the capacity of UTMs? What would that be? Also, the Church-Turing-thesis is in some sense a definition of computation. Does the thesis appeal to something like hypercomputation?
Sec. 5.2 “Agency-based hierarchies of levels for an autonomous agent”
How are the agency-based account of information and the “information is everywhere” account related? On the one hand, much of the statements derive from the success of interpreting physiological processes as a result of the action of semi-autonomous agents. On the other hand, GDC speculates repeatedly about the fundamental physical nature of information. Are agents fundamental entities at one level? Why do you need the connection to fundamental physics at all to make your case?
It seems to me that MM objects specifically to the claim that “Information is Everywhere” (objection #(1)), but I failed to see why GDC appealed to it in the first place.
Verdict
I think the article is interesting, but it is also a bit superficial/general at times. There is a flurry of points and concepts which are not really extensively argued for but apparently play a role in the larger argument (what is it, I still haven’t found out?) The article, in 22 pages (!), goes into three very substantial debates about ontology, computation, and cognition. I am not sure how these points (information = everywhere, computation = physical information processing != what UTMs do, cognition = computation) work together, and I was slightly confused after reading it thinking “so what?” In addition, I found the structure a bit confusing.
I would therefore recommend "revise and resubmit".
Author Response
Please find our replies attached.

Reviewer 2 Report (New Reviewer)
Discussion on the relationship between computation, information, cognition, and their embodiment is the title of this interesting and very rich paper, written by two authors that are productively active in this area of study. The problem is to enhance our knowledge of the concept of information, and the paper shows that this can be reached by deepening the intellectual interaction with the concepts of cognition and computation. The paper also takes advantage of a dialectical discussion that I had the opportunity of appreciating because of the richness of references also to recent results in the area of cognitive science.
Author Response
Thank you for the assessment of our paper.
Round 2
Reviewer 1 Report (New Reviewer)
The authors addressed all my points, and I recommend publishing this piece. It is non-technical in nature, but I defer to the editors of the SI to evaluate its appropriateness for Entropy.
I found the restructuring and the additions quite helpful. Also, the fact that the "summary of the argument" changed into "definitions of terms" is good.
As to the point on UTM: I think this is now mainly about a review-internal distinction between "not in principle computable by UTM but by biological architectures" and "not in practice computable by a TM ..." (for reasons given in 2.4). I would endorse the latter and remain skeptical of the former. (But the former is not explicitly said in the paper, and it would probably need a paper on its own. But this is not thinpoint.)
Also re my previous comment on the "flurry of ideas": the author somewhat agreed with this statement, but also gave a good reason why one should sometimes nevertheless start with a flurry of seemingly unconnected ideas and that it's difficult to get on with the job of synthesizing something interesting (rather than only trying to defend the individual theses). I remain somewhat unconvinced that "something interesting" indeed has been synthesized, but this is for the readers to judge for themselves. As a referee, I am happy with the article as is.
This manuscript is a resubmission of an earlier submission. The following is a list of the peer review reports and author responses from that submission.
Round 1
Reviewer 1 Report
The article is not at all scientific, this can be easily seen by reading the Introduction. There is no standard statement of the problem with the definition of the corresponding goal and methods for achieving it. The background of the study is not specified. In addition to this, we see some very common phrases with only one reference. The main body of the article contains some very naive statements with references to very popular names such as Darwin or Shannon, without demonstrating that the authors have read anything recent in the field. It seems to me that the article was written by an undergraduate student who does not know how scientific papers should be written. The paper must be immideately rejected without reconsideration.
Reviewer 2 Report
The manuscript is a nice discussion on extremely important topics, namely computation, information and cognition. It raises many intriguing and interesting points and it provides a wealth of relevant references.
Nevertheless, in my view it is a perfect final chapter of conference proceedings, rather than a journal paper (even though a "perspective" one).
The main concepts and issues are often just sketched, pointing to references. Very often the reader is disappointed by observing that some concepts are not delved but just quickly mentioned. Moreover, while I think that the subject is extremely relevant and interesting, I frankly don't see such a depth in the conclusions, which seem more a way for finding a respectful agreement between the two discussants.
If the authors want to pursue the way of preparing a journal article, my suggestion is to write a lengthy and detailed review paper, which elaborates the main concepts and issues in depth. A large part of the current manuscript can be used as a skeleton for such a work and the discussion can be an excellent appendix. But, in my opinion, the current manuscript cannot be accepted for publication.
Reviewer 3 Report
Please the attachment.
